# Population pharmacokinetics of DNDI-6148 in healthy adults

Frauke Assmus[1,2], Ayorinde Adehin[1,2], Richard M. Hoglund[1,2], Charles E. Mowbray[3], Jean-Yves Gillon[3], Séverine Blesson[3], Stéphanie Braillard[3], Eric Chatelain[3], Ivan Scandale[3], Joel Tarning[1,2,4]*

1 Mahidol Oxford Tropical Medicine Research Unit, Faculty of Tropical Medicine, Mahidol University, Bangkok, Thailand, 2 Centre for Tropical Medicine and Global Health, Nuffield Department of Medicine, University of Oxford, Oxford, United Kingdom, 3 Drugs for Neglected Disease initiative, Geneva, Switzerland, 4 Infectious Diseases Data Observatory, Oxford, United Kingdom

* joel.tarning@ndm.ox.ac.uk

## Abstract

Leishmaniasis (both visceral and cutaneous) and Chagas disease (CD) are among the most neglected tropical diseases, with limited treatment options and an urgent need for safer, more effective therapies. DNDI-6148, a benzoxaborole with anti-leishmanial and antichagasic activity, is currently under clinical development. A first-in-human (FIH), Phase 1 study (ISRCTN54981564) has recently assessed the safety, tolerability, and pharmacokinetics of DNDI-6148, demonstrating a good safety profile and, based on non-compartmental analysis, non-linear pharmacokinetics. To support dose selection for future clinical trials, we conducted a population pharmacokinetic analysis using data from the FIH study. The analysis included data from 48 healthy male participants who received a single oral dose of DNDI-6148 (10–380 mg) across eight dosing cohorts. Plasma concentrations were quantified by liquid chromatography–tandem mass spectrometry, and concentration–time data were pooled and analyzed using nonlinear mixed-effects modeling. DNDI-6148 pharmacokinetics were non-linear and best described by a one-compartment disposition model. Higher doses were associated with decreased relative bioavailability and clearance, resulting in less than dose-proportional increases in peak plasma concentrations. The median elimination half-life increased with dose, ranging from 12.6 to 33.7 hours. In summary, the population pharmacokinetic model adequately described DNDI-6148 pharmacokinetics in healthy participants. It provides a valuable tool to guide dose selection for future clinical trials in patients with leishmaniasis and Chagas disease.

## Author summary

Chagas disease and leishmaniasis are neglected tropical diseases that disproportionately affect the world's poorest populations, causing substantial morbidity

---

---

**Data availability statement:** The data underlying the results of this study are available upon request because they contain potentially sensitive personal information, which must be de-identified at the individual level. Interested researchers may request access to de-identified participant data from Vivli, the data-sharing partner of the Drugs for Neglected Diseases initiative (DNDi), commissioner of this study, at https://vivli.org/ourmember/dndi/.

**Funding:** This research was funded in part by the Wellcome Trust (220211, granted to JT, RMH) and DNDi. For these activities, DNDi received funding from UK Aid, UK; the Swiss Agency for Development and Cooperation (SDC); the Dutch Ministry of Foreign Affairs (DGIS), the Netherlands; and for its overall mission, from Médecins Sans Frontières, International. The funders had no role in the study design, data collection and analysis, decision to publish, or preparation of the manuscript. For the purpose of open access, the author has applied a CC BY public copyright license to any Author Accepted Manuscript version arising from this submission.

**Competing interests:** The authors have declared that no competing interests exist.

and mortality in affected regions. Current treatments for these infectious diseases face major challenges, including long treatment durations, variable efficacy, safety concerns, high cost, and logistical barriers to access. Consequently, there is an urgent need for new, safer, and more accessible oral therapies. DNDI-6148 is a novel drug candidate with potent antiparasitic activity against *Trypanosoma cruzi* and multiple *Leishmania* species, as shown in laboratory experiments and preclinical animal studies. The drug has recently been administered to healthy adults for the first time and was well tolerated. In this study, we analyzed data from this first-in-human clinical trial to better understand the drug's pharmacokinetic behavior after oral dosing. Using a modeling approach, we described the relationship between dose and drug exposure and showed that DNDI-6148 displays non-linear pharmacokinetic behavior, meaning that drug exposure in plasma does not increase proportionally with dose. By characterizing these relationships, our work provides a quantitative framework to guide dose selection in future clinical trials. These findings represent an important step toward developing an effective and accessible oral treatment for Chagas disease and leishmaniasis.

## Introduction

Leishmaniasis and Chagas disease (CD) are infectious diseases caused by protozoan parasites of the *Trypanosomatidae* family [1–3]. They remain major public health challenges in low- and middle-income countries, where they are closely linked to poverty and cause substantial morbidity and mortality [4–6]. Although classified as neglected tropical diseases (NTDs) [7], their distribution extends beyond tropical regions [8, 9].

Leishmaniases are a complex of parasitic diseases caused by more than 20 *Leishmania* species, transmitted through the bite of infected female sandflies [2]. The highest disease burden is found in Latin America, South-East Asia, East Africa, and North Africa [10], with 700 000–1 million new cases annually worldwide [2]. Leishmaniasis manifests in various forms, from localized, self-healing skin lesions to potentially fatal systemic disease, depending on the *Leishmania* strain and host immune response [3, 11, 12]. Cutaneous leishmaniasis (CL) is the most common form [2, 13], while visceral leishmaniasis (VL) is the most severe and almost always fatal if left untreated [2]. CD, caused by *Trypanosoma cruzi* (*T. cruzi*) and primarily transmitted by infected triatomine bugs, is endemic in 21 Latin American countries, with an estimated 6–7 million people infected [1]. Chronic *T. cruzi* infection can lead to life-threatening complications, particularly cardiomyopathy and, less frequently, gastrointestinal disorders, affecting about one-third of chronically infected individuals [1, 9, 14].

Despite the high disease burden, treatment options for CL, VL, and CD are limited and far from ideal [15–17]. Leishmaniasis treatment varies by region, parasite species, clinical manifestation, and drug availability [18, 19]. Most therapies require parenteral administration and face issues of safety, logistical barriers, and regional

differences in efficacy [20]. Miltefosine, the only oral drug for leishmaniasis, was a major advance but is contraindicated in pregnancy and often poorly tolerated due to gastrointestinal adverse effects, particularly vomiting, which can reduce treatment compliance. It is also challenged by emerging resistance and strain-specific efficacy [21–23]. For CD, the only available dugs (benznidazole, nifurtimox) were developed over 50 years ago and have poor tolerability, variable efficacy in the chronic phase and long treatment durations [24–27]. In the absence of vaccines, these limitations underscore the urgent need for safer, more effective, and accessible treatments for leishmaniasis and CD [10, 28–30].

DNDI-6148 is a novel benzoxaborole derivative with broad antiparasitic activity [31, 32]. It has shown potent *in vitro* activity against multiple *Leishmania* species (e.g., *L. donovani, L. infantum, L. major, L. tropica*) and *T. cruzi* [31, 32], making it a strong candidate for both leishmaniasis and CD. In preclinical studies, it achieved >95% parasite burden reduction in mouse and hamster models of VL, with similar efficacy in acute and chronic infection models [31, 33]. It also significantly reduced parasite load and lesion size in *L. major*–BALB/c mouse models of CL [32]. In *Leishmania spp.*, DNDI-6148 inhibits the cleavage and polyadenylation specificity factor (CPSF3) endonuclease, a critical enzyme for mRNA processing and parasite survival [31]. CPSF3 has been identified as a conserved benzoxaborole target across multiple parasites, including *T. cruzi* and *Leishmania spp.* [34, 35]. Although the specific target of DNDI-6148 in *T. cruzi* has not yet been fully characterized, it is likely to involve the same mechanism [36]. Preclinical studies established oral bioavailability and supported advancement into clinical trials [31].

Recently, the safety, tolerability, and pharmacokinetics (PK) of DNDI-6148 were evaluated in healthy male participants following single ascending oral doses up to 380 mg [37]. DNDI-6148 was well tolerated in this first-in-human (FIH) study and demonstrated a favourable PK profile. Non-compartmental analysis revealed slow absorption, nonlinear pharmacokinetics and only trace amounts of three metabolites in plasma. To better quantify the dose–exposure relationship and provide a foundation for future dose-finding studies, we conducted a population pharmacokinetic analysis of DNDI-6148 using data from this FIH trial.

## Methods

### Clinical data

**Ethics statement.** The study was approved by the Comité de Protection des Personnes Sud-Ouest et Outre-Mer II and authorized by the French regulatory agency (ANSM). It was conducted in accordance with the Declaration of Helsinki and ICH-GCP (E6) guidelines. All participants provided written informed consent before any trial related investigations.

**Study design.** This population PK analysis was based on data from a Phase 1 FIH study of DNDI-6148 (EudraCT 2018-004023-37; ISRCTN54981564), conducted between 2018 and 2022 at Eurofins-Optimed (Gières, France). Full study details have been published previously [37]; a brief summary is provided below.

This was a randomized, double-blind, placebo-controlled, single-center, single ascending dose (SAD) study. A total of 64 healthy Caucasian men aged 18–50 years were enrolled. Participants received a single oral dose of DNDI-6148 or placebo under fasting conditions. The study drug was supplied as a powder for suspension (DNDI-6148 as the arginine monohydrate), reconstituted in ORA-Sweet vehicle (Perrigo, Minneapolis, MN, USA). Doses ranged from 10 to 380 mg across eight cohorts (10, 20, 40, 80, 160, 220, 300, and 380 mg; free acid equivalent). Each cohort included 8 subjects, with 6 receiving the active drug and 2 receiving placebo.

**Pharmacokinetic sample collection and drug quantification.** Blood samples were collected from all subjects at specified time points to quantify DNDI-6148 plasma concentrations. On Day 1, blood was drawn via catheter; on subsequent days, single-use needles were used. Sampling schedules varied by cohort. For subjects receiving 10–80 mg, samples were collected pre-dose and at 0.5, 1, 1.5, 2, 2.5, 3, 4, 6, 9, 12, 24, 48, 72 hours post-dose. For those receiving 160–380 mg, samples were collected pre-dose and at 0.5, 1, 2, 2.5, 3, 4, 5, 6, 9, 12, 24, 48, 72, 96, and 120 hours post-dose. For the highest dose groups (300 and 380 mg), an additional sample was collected at 168 hours post-dose. Exact sampling times were recorded for use in population PK modeling.

Plasma concentrations of DNDI-6148 in acidified $K_2$EDTA plasma were quantified using a validated, internally standardized liquid chromatography–tandem mass spectrometry (LC–MS/MS) method. The lower limit of quantification (LLOQ) was 1 ng/mL. Between-run and within-run assay precision were 10.10% and 4.63% CV (coefficient of variation), respectively, both observed at the LLOQ. Bioanalysis was conducted at SGS Life Sciences (Wavre, Belgium) in accordance with Good Laboratory Practice (GLP).

## Population pharmacokinetic analysis

### (i) Model development

Plasma concentrations of DNDI-6148 were pooled across all treatment arms (48 subjects) and transformed using the natural logarithm. All available PK samples were included in the analysis. Concentration-time profiles were modeled simultaneously using nonlinear mixed-effects modeling in NONMEM (v7.4.3; Icon Development Solution, Ellicott City, MD, USA), employing the first-order conditional estimation method with interactions (FOCE-I). Model building and diagnostics were supported by Pirana (v2.9.9), Pearl-speaks-NONMEM (PsN v5.2), R (v4.2.2), and TIBCO Spotfire (v11.3.0).

### (ii) Structural and stochastic model

One- and two-compartment disposition models were evaluated with first-order absorption, with or without lag time. Transit compartment models comprising one to five transit compartments were also assessed, with the absorption rate constant ($K_A$) and transit rate constant ($K_{TR}$) either fixed to equal values or estimated separately.

Relative bioavailability (F) was fixed to unity for the population, with an estimated inter-individual variability (IIV) incorporated into the base model. IIV was implemented using an exponential error model. Residual unexplained variability was implemented as an additive error on log-transformed concentrations, equivalent to an exponential error on the arithmetic scale.

### (iii) Covariate model

Covariate effects were explored based on biological plausibility and statistical significance. Body weight (standardized to 70 kg) was implemented a priori as an allometric function on all clearance (exponent 0.75) and volume (exponent 1) parameters. Following inclusion of allometric scaling, other covariate effects were explored using a stepwise covariate modelling (SCM) approach. For all covariates, linear, exponential, and power relationships centered on population median values were evaluated in parallel using the *scm* functionality in PsN. Additional investigated covariates included dose (mg/kg), demographic and laboratory parameters (age, markers of liver and kidney function, hematocrit), as well as creatinine clearance (calculated from plasma creatinine levels using the Cockcroft–Gault equation). In addition, the impact of dose (mg/kg) on F and clearance (CL) was also evaluated with the use of saturation models (i.e., Emax-type models). Covariate selection was guided by changes in the objective function value (OFV), with forward selection at $p < 0.05$ ($\Delta OFV \geq 3.84$) and backward elimination at $p < 0.001$ ($\Delta OFV \geq 10.83$).

### (iv) Model evaluation

Model selection was based on the OFV, with a reduction in $OFV \geq 3.84$ indicating a significant improvement between nested models ($p < 0.05$, one degree of freedom). Model performance was evaluated using goodness-of-fit (GOF) plots, prediction-corrected visual predictive checks (VPCs, n = 1000), and bootstrap analysis (n = 1000) to estimate parameter precision (relative standard errors [RSE%], and 95% confidence intervals [CI]).

Secondary PK parameters, including time to maximum concentration ($T_{MAX}$), peak concentration ($C_{MAX}$), and area under the concentration-time curve extrapolated to infinity ($AUC_\infty$), were calculated along with dose-normalized values.

## Results

### Study population

This Phase 1 SAD study enrolled 64 healthy male participants across eight dose cohorts (10 – 380 mg DNDI-6148, free acid equivalent). Participants received a single oral dose of DNDI-6148 arginine monohydrate (n = 48) or placebo (n = 16) as an oral suspension under fasting conditions. All participants completed the study.

Demographic and baseline characteristics of all 64 participants have been reported previously [37]. Key characteristics of the participants included in the population PK analysis are summarized in Table 1. All were healthy men of White ethnicity, aged 18–50 years, with a median body weight of 72.3 kg. Baseline characteristics appeared comparable between dose groups.

### Pharmacokinetic data

The PK analysis included all 48 subjects who received DNDI-6148, contributing a total of 684 plasma concentration samples. All post-dose samples were above the lower limit of quantification (LLOQ).

**Table 1. Demographic and baseline laboratory characteristics of the subjects participating in the DNDI-6148 FIH study.**

| Characteristic | 10 mg (n=6) | 20 mg (n=6) | 40 mg (n=6) | 80 mg (n=6) | 160 mg (n=6) | 220 mg (n=6) | 300 mg (n=6) | 380 mg (n=6) | Pooled PK group (n=48) |
|---|---|---|---|---|---|---|---|---|---|
| Dose (mg/kg) | 0.14 (0.12–0.17) | 0.31 (0.28–0.33) | 0.57 (0.45–0.61) | 1.03 (0.95–1.22) | 2.26 (2.19–2.42) | 3.16 (2.93–3.87) | 3.55 (3.11–4.26) | 4.53 (4.14–5.56) | 1.70 (0.12–5.56) |
| Sex (male %) | 100 | 100 | 100 | 100 | 100 | 100 | 100 | 100 | 100 |
| Age (years) | 37.5 (21–43) | 23 (19–40) | 47 (33–50) | 42.5 (28–48) | 33 (18–48) | 25 (18–46) | 33.5 (18–50) | 39 (28–49) | 35 (18–50) |
| Weight (kg) | 69.4 (60.0–84.5) | 65.3 (59.9–72.5) | 70.5 (65.2–89.0) | 78 (65.7–84.1) | 70.7 (66.2–73.2) | 69.6 (56.9–75.2) | 84.4 (70.5–96.5) | 84.0 (68.4–91.8) | 72.3 (56.9–96.5) |
| Body mass index (kg/m²) | 23.9 (19.7–27.3) | 21.6 (18.5–24.2) | 24.1 (20.8–28.2) | 23.2 (21.7–29.1) | 22.6 (21.2–23.2) | 22.1 (19.7–22.8) | 26.2 (22.0–29.8) | 27.5 (19.8–29.8) | 22.9 (18.5–29.8) |
| Aspartate Aminotransferase (IU/L) | 22 (13–33) | 19 (13–29) | 19 (15–22) | 20.5 (16–24) | 18 (13–29) | 20 (10–32) | 19 (14–23) | 21 (18–30) | 19 (10–33) |
| Alanine Aminotransferase (IU/L) | 22 (13–40) | 16 (12–25) | 15 (13–19) | 18.5 (12–30) | 16 (7–22) | 24 (11–40) | 22 (16–25) | 21.5 (18–35) | 19 (7–40) |
| Alkaline Phosphatase (IU/L) | 55 (49–113) | 58 (36–77) | 57 (49–94) | 62 (48–78) | 72 (48–102) | 74.5 (64–101) | 71.5 (50–95) | 64 (58–77) | 66.5 (36–113) |
| Bilirubin (µmol/L) | 10.5 (5–17) | 14.5 (3–26) | 9 (3–13) | 11 (7–13) | 10 (7–24) | 9 (5–22) | 8 (3–10) | 11 (9–19) | 10 (3–26) |
| Gamma-glutamyl transferase (IU/L) | 13 (9–51) | 13.5 (10–19) | 17 (11–41) | 15.5 (8–22) | 15 (10–19) | 16.5 (10–36) | 26.5 (9–39) | 15.5 (11–37) | 15.5 (8–51) |
| Creatinine Clearance (mL/min) | 101 (91.5–133) | 108.3 (85.4–119) | 110 (95.3–123) | 95.6 (81.5–124) | 103 (85.1–130) | 127 (90.9–152) | 146 (88.5–161) | 118 (112–129) | 113 (81.5–161) |
| Creatine Phosphokinase (IU/L) | 86.5 (56–209) | 97 (79–134) | 81.5 (67–170) | 110.5 (85–128) | 109.5 (84–172) | 82.5 (63–186) | 94 (72–202) | 155 (93–303) | 98.5 (56–303) |
| Hematocrit (%) | 43.1 (39.5–46.3) | 45.1 (41.5–48.4) | 41.3 (38.5–43.5) | 41.5 (37.1–44.8) | 42.9 (40.9–47.7) | 45.5 (43.7–46.7) | 44.4 (41.8–47.5) | 42.6 (40.0–44.0) | 43.0 (37.1–48.4) |

**All participants were of White ethnicity.** All values are given as median (minimum - maximum range), except sex (% of subjects). **Abbreviations:** PK, Pharmacokinetic.

Fig 1 shows individual concentration–time profiles by dose group, with an overlay of median concentrations shown in the bottom-right panel. Two post-dose samples—one scheduled at 1 h and one at 120 h, each from a different participant—deviated by more than 10% from the scheduled sampling time per protocol, but this had minimal impact on the medians. All values were retained in the population PK analysis, which used actual recorded sampling times. The analysis was based solely on plasma concentrations of DNDI-6148, the predominant circulating species [37].

## Population pharmacokinetic model

The plasma concentration-time profile of DNDI-6148 was best described by a one-compartment disposition model with first-order absorption. A two-compartment model was tested but was not carried forward due to low precision in additional PK parameter estimates (> 300% RSE). More complex absorption models, including transit compartment absorption models, were also evaluated but did not significantly improve the model fit.

Body weight was included as an allometric function on apparent clearance (CL/F) and apparent volume of distribution (V/F), resulting in a model improvement (ΔOFV = -8.8). In addition, dose (mg/kg) was identified as a significant covariate on relative bioavailability (F) and subsequently on CL/F. A dose effect on F alone was insufficient to capture the observed

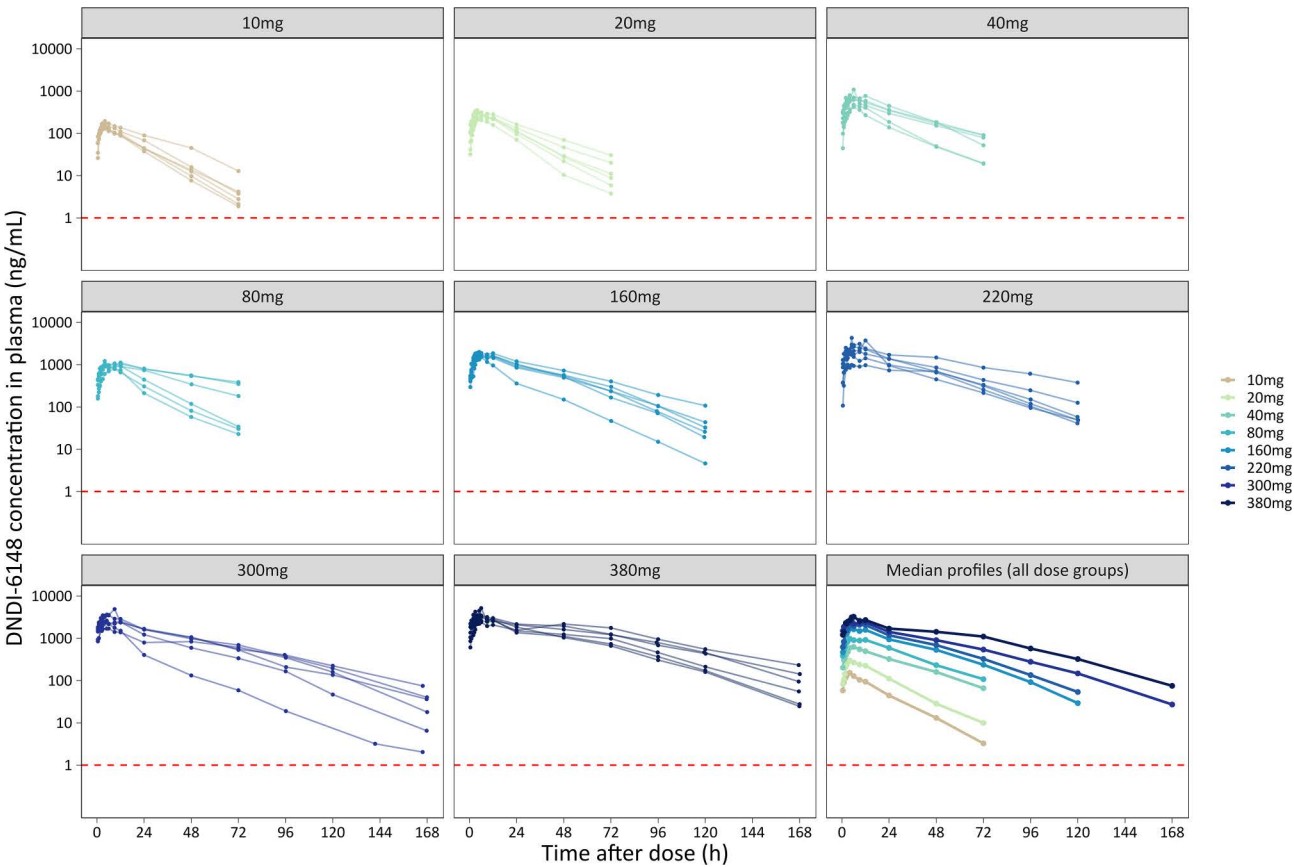

**Fig 1. Plasma concentration–time profiles of DNDI-6148 following single oral doses of 10–380 mg.** Each panel displays individual PK profiles stratified by dose group. The bottom-right panel shows median concentrations at each nominal time point across all dose groups. The horizontal red dashed line indicates the lower limit of quantification (1 ng/mL).

increase in terminal elimination half-life at higher doses and approximately dose-proportional $AUC_\infty$, necessitating the inclusion of an additional dose effect on CL/F.

For the retained dose effects on F and CL/F, power functions resulted in larger reductions in OFV but produced steep relationships at low doses. Exponential functions centered on the median dose ($Dose_{median} = 1.7$ mg/kg) were therefore selected for the final model as a more conservative approach with improved suitability for translational applications. Higher doses were associated with decreases in both F ($\Delta OFV = -28.0$) and CL/F ($\Delta OFV = -23.0$). These dose effects were ultimately implemented as follows:

$$F_i = F \times e^{\theta_{Dose\_F} \times (Dose_i - Dose_{median})}$$
(Eq.1)

$$CL_i/F_i = CL/F \times e^{\theta_{Dose\_CL} \times (Dose_i - Dose_{median})}$$
(Eq.2)

where $F_i$ and $CL/F_i$ represent the individual relative bioavailability and apparent clearance at a given dose ($Dose_i$, mg/kg). The coefficients $\Theta_{Dose\_F}$ and $\Theta_{Dose\_CL}$ are the estimated coefficients for the dose effects. Based on these estimates, F decreased from 121% at the lowest dose (0.12 mg/kg) to 100% at the median and 62% at the highest dose (5.56 mg/kg). Similarly, CL/F decreased from 3.2 L/h at the lowest dose to 2.55 L/h at the median and 1.4 L/h at the highest dose. To aid interpretation of the implemented dose effects, S1 Fig illustrates the exponential relationships of F and CL/F as functions of dose used in the final model.

Alternative Emax models were explored for the dose effects on F and CL/F and yielded a marginally better fit than the exponential model ($\Delta OFV = -3.24$). However, the additional parameters were estimated with high imprecision (>120% RSE for $D_{50}$ values), and the improvement in model fit was not sufficient to justify the added complexity. For parsimony, the exponential functions were retained. No other covariates tested during model development were retained after backward elimination.

Parameter estimates from the final population PK model are summarized in Table 2. The absorption rate constant ($K_A$) exhibited high inter-individual variability (56% CV), while clearance showed moderate variability (31% CV). Inter-individual variability on apparent volume of distribution was estimated to be close to zero and was therefore fixed to zero in the final model. All parameter estimates were precise, with RSE values < 15%, as confirmed by bootstrap analysis.

**Table 2. Parameter estimates of the final population PK model of DNDI-6148.**

| Parameter | Population estimate[a] (%RSE)[b] | Bootstrapping 95% CI[b] | IIV, %CV[a] (%RSE)[b] | Bootstrapping 95% CI[b] |
|---|---|---|---|---|
| Relative oral bioavailability, F | 1 *fixed* | – | 17.0 (13.6) | 11.9, 20.9 |
| Absorption rate constant, $K_A$ (h⁻¹) | 0.576 (7.7) | 0.497, 0.667 | 56.1 (10.9) | 44.1, 67.2 |
| Apparent clearance, CL/F (L/h) | 2.55 (5.9) | 2.28, 2.85 | 30.8 (12.8) | 23.2, 38.2 |
| Apparent volume of distribution, V/F (L) | 69.9 (2.8) | 66.4, 73.9 | – | – |
| $\theta_{Dose\_F}$ (dose effect on F)[c] | -0.123 (13.5) | -0.158, -0.094 | – | – |
| $\theta_{Dose\_CL}$ (dose effect on CL/F)[c] | -0.150 (12.8) | -0.201, -0.096 | – | – |
| Variance of residual error, σ | 0.0361 (13.4) | 0.0271, 0.0460 | – | – |

Population estimates are given for an adult weighting 70 kg.

[a] Population mean parameter estimates and IIV calculated by NONMEM. The coefficient of variation (% CV) for IIV was calculated as $100 \times \sqrt{e^{\omega^2} - 1}$.

[b] Precision of parameter estimates, based on nonparametric bootstrap diagnostics of the final PK model. RSEs (%) are calculated as $100 \times \frac{standard\ deviation}{mean\ value}$. The 95% CIs are based on the 2.5th –97.5th percentiles of the bootstrap parameter estimates.

[c] Dose effects on F and CL/F were described using exponential functions centered on the median dose (1.7 mg/kg): $F_i = F \times e^{\theta_{Dose\_F} \times (Dose_i - Dose_{median})}$ and $CL_i/F_i = CL/F \times e^{\theta_{Dose\_CL} \times (Dose_i - Dose_{median})}$, where $Dose_i$ is the individual dose.

The final model adequately described the observed concentration-time profiles, with no major model misspecifications and good predictive performance. GOF plots and the prediction-corrected VPC are shown in Figs 2,3, respectively. The NONMEM code for the final PK model is provided as S1 Code.

Summary statistics of secondary PK parameter estimates for DNDI-6148, as derived from the population PK model, are presented in Table 3. DNDI-6148 was slowly absorbed, with median $T_{MAX}$ values ranging from 3.5 to 6.1 hours across dose cohorts (individual range: 2.2 - 11.2 hours). The median elimination half-life ($t_{1/2}$) was approximately 13 hours for the 10 and 20 mg doses, increasing to ~20 hours for the 40–220 mg dose groups, and reaching 33.7 hours at the highest dose. Individual $t_{1/2}$ values ranged from 10.8 to 50.0 hours.

$C_{MAX}$ increased with dose but showed less-than-dose-proportional behavior, particularly above 80 mg. A 38-fold increase in dose (from 10 mg to 380 mg) resulted in only a 22-fold increase in $C_{MAX}$, consistent with the dose-dependent reduction in relative bioavailability. Distributions of individual primary and secondary PK parameters by dose group are shown in S2 Fig, highlighting the dose-dependent trends and variability.

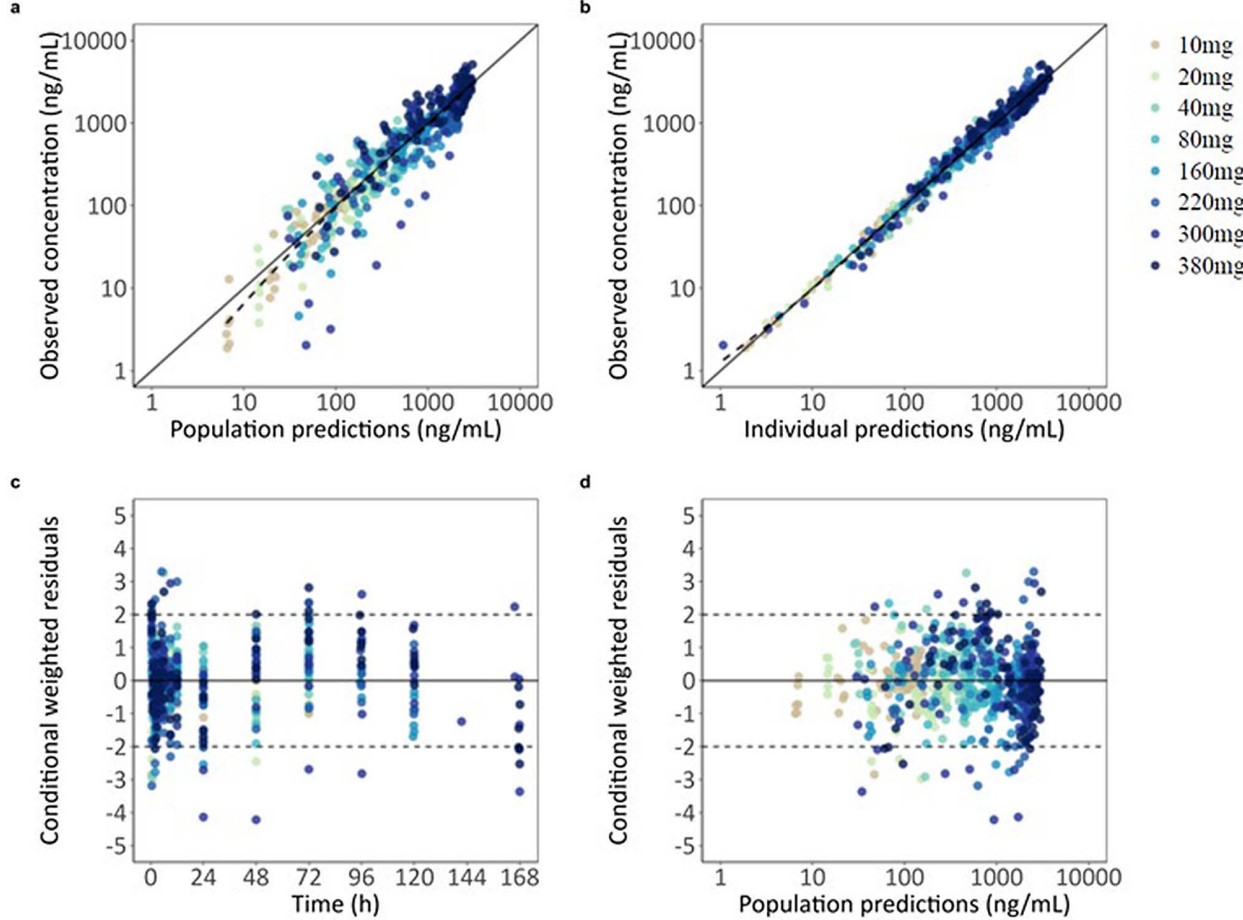

**Fig 2. Goodness-of-fit of the final population PK model for DNDI-6148. a)** observed versus population predicted concentrations, **b)** observed versus individually predicted concentrations, c) conditionally weighted residuals versus time, and **d)** conditionally weighted residuals versus population predicted concentrations. Observations are represented by circles. Solid black lines represent the line of identity **(a, b)** or zero line **(c, d)**. Dashed lines indicate LOESS smoothing of the data (a, b).

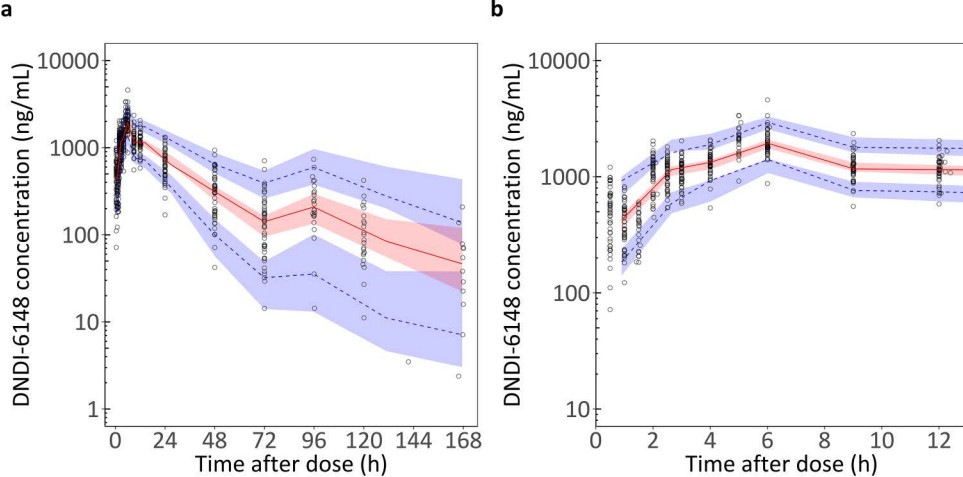

**Fig 3. Prediction-corrected visual predictive check of the final population PK model for DNDI-6148. a)** Full sampling period; **b)** zoom into the absorption phase (0–12 h). Open circles represent observed DNDI-6148 concentrations in plasma. The solid red line shows the median (50th percentile) of the observed data; while the blue dashed lines represent the 5th and 95th percentiles. Shaded areas represent the 95% confidence intervals for the corresponding simulated percentiles (5th, 50th, and 95th).

**Table 3. Summary of key pharmacokinetic parameters for DNDI-6148 in plasma.**

| Parameter | 10 mg | 20 mg | 40 mg | 80 mg | 160 mg | 220 mg | 300 mg | 380 mg |
|---|---|---|---|---|---|---|---|---|
| $T_{MAX}$ (h) | 4.23 (2.89 - 4.64) | 5.23 (3.66 - 8.89) | 5.54 (3.78 - 7.23) | 5.34 (3.81 - 9.61) | 6.13 (5.42 - 7.17) | 5.69 (3.73 - 11.20) | 3.50 (2.53 - 6.00) | 5.91 (3.29 - 8.77) |
| $C_{MAX}$ (ng/mL) | 138 (130 - 169) | 265 (230 - 306) | 590 (368 - 700) | 1,007 (714 - 1,071) | 1,792 (1,272 - 1,870) | 2,184 (1,428 - 2,512) | 2,459 (2,007 - 3,360) | 3,088 (2,638 - 3,607) |
| $AUC_{\infty}$ (µg × h/mL) | 3.05 (2.60 - 5.00) | 7.15 (4.95 - 9.25) | 21.4 (10.4 - 25.5) | 34.4 (17.7 - 63.5) | 61.7 (38.5 - 81.0) | 86.9 (61.6 -149) | 113 (60.7 -126) | 159 (135 - 206) |
| $t_{1/2}$ (h) | 12.6 (10.9 - 16.9) | 13.2 (10.9 - 18.1) | 20.7 (13.6 - 23.0) | 19.8 (12.5 - 36.4) | 19.9 (14.9 - 25.7) | 22.5 (19.5 - 35.7) | 26.6 (15.8 - 31.9) | 33.7 (24.6 - 46.9) |
| Cl/F (L/h) | 4.01 (2.69 - 4.90) | 3.27 (2.65 - 4.22) | 2.60 (2.27 - 3.54) | 2.70 (1.47 - 4.59) | 2.41 (1.85 - 3.42) | 2.17 (1.34 - 2.31) | 2.24 (2.05 - 3.27) | 1.74 (1.26 - 2.47) |
| V /F (L) | 69.3 (60.9 - 83.2) | 65.3 (60.4 - 70.9) | 70.5 (65.8 - 86.6) | 77.9 (66.1 - 83.8) | 70.6 (66.3 - 72.9) | 69.6 (59.6 - 74.6) | 84.4 (71.5 - 95.1) | 84.0 (69.6 - 91.6) |
| F | 1.29 (1.07 - 1.38) | 1.17 (0.99 - 1.30) | 1.30 (0.90 - 1.50) | 1.09 (1.01 - 1.21) | 0.92 (0.75 - 1.02) | 0.75 (0.63 - 0.94) | 0.86 (0.61 - 0.95) | 0.70 (0.67 - 0.93) |

All values are given as median (5th to 95th percentile). **Abbreviations:** $T_{MAX}$, time to maximum concentration; $C_{MAX}$, peak plasma concentrations; $AUC_{\infty}$, area under the concentration-time curves to infinity; V/F, apparent volume of distribution; CL/F, apparent clearance; $t_{1/2}$, half-life.

In contrast, $AUC_{\infty}$ increased approximately dose-linearly, likely due to dose-dependent decrease in apparent clearance (CL/F) that offset the reduced bioavailability at higher doses. Fig 4 shows dose-normalized $C_{MAX}$ and $AUC_{\infty}$ values versus dose, illustrating nonlinearity for $C_{MAX}$ and approximate dose-proportionality for $AUC_{\infty}$.

## Discussion

Leishmaniasis and CD disproportionately affect the world's poorest populations and are closely linked to global health inequity due to limited treatment options and poor access to care. New oral therapies are urgently needed to improve patient outcomes and achieve WHO targets [10, 28–30, 38].

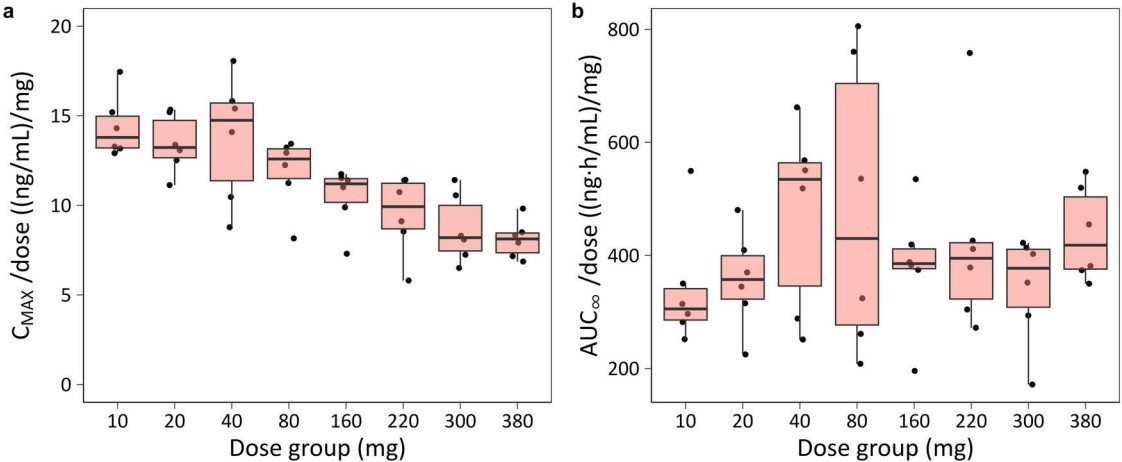

**Fig 4. Dose-normalized plasma exposure of DNDI-6148 by dose groups. a)** $C_{MAX}$/dose; **b)** $AUC_\infty$/dose. Boxes represent the interquartile range (IQR) with median (line); whiskers extend to $1.5 \times IQR$, and dots indicate individual observations.

Benzoxaboroles have gained increasing attention for their unique pharmacological potential, including broad antiparasitic activity and high target specificity [39]. Notably, acoziborole has demonstrated excellent efficacy as a single-dose oral treatment for human African trypanosomiasis (HAT) in a pivotal phase 2/3 trial [40, 41], highlighting the promise of this chemical class.

DNDI-6148, a novel benzoxaborole currently under clinical development, has shown potent antiparasitic activity against multiple *Leishmania* species and *T. cruzi* [31, 32]. This study presents the first population PK analysis of DNDI-6148 in humans providing key insights into its pharmacokinetic properties and supporting its continued drug development.

## Pharmacokinetic properties of DNDI-6148

DNDI-6148 showed relatively slow absorption (median $T_{MAX}$: 3.5 to 6.1 hours across cohorts) and dose-dependent nonlinear pharmacokinetics, consistent with the previously published non-compartmental analysis [37]. While $AUC_\infty$ increased approximately dose-linearly, $C_{MAX}$ rose less than dose-proportionally, particularly at doses >80 mg. This pattern suggests a dose-dependent reduction in relative bioavailability, that appeared to be offset by a concurrent decrease in elimination clearance associated with higher doses. Although intravenous data were unavailable to determine absolute bioavailability, systemic exposure was substantial, with a median $C_{MAX}$ of 3,088 ng/mL at the 380 mg dose.

DNDI-6148 has low-to-moderate solubility in physiologically relevant media [31]. To enhance solubility, an arginine salt formulation was administered as an oral suspension stabilized in ORA-Sweet under fasting conditions. Permeability in Caco-2 cells was reported as good with no evidence of P-glycoprotein-mediated efflux [31, 42], suggesting that poor solubility – rather than permeability – limits absorption at higher doses. The dose-dependent decline in bioavailability highlights the potential to improve systemic exposure through alternative formulations with enhanced solubility. Future research should explore such formulations in later-phase trials to support clinical development.

The disposition of DNDI-6148 was well captured by a one-compartment model (V/F ~ 70 L; CL/F = 2.55 L/h). The moderate volume of distribution aligns with DNDI-6148's low-to-moderate lipophilicity (logD 1.92 at pH 7.4) and moderate plasma protein binding (92% bound to human plasma) [31].

Total clearance was low (2.55 L/h at the median dose), consistent with preclinical data in rats, although higher clearance was observed in dogs [31]. The apparent dose-dependent reduction in clearance may reflect saturation of a non-renal elimination pathway, potentially involving hepatic metabolism or biliary excretion. The latter is likely

predominant, given the low levels of circulating metabolites, and would be consistent with the slow biliary–fecal elimination reported for acoziborole in a human mass balance study [43]. Renal excretion was minimal, with < 0.2% of the unchanged drug recovered in urine [36]. Metabolite profiling detected only trace levels of circulating metabolites, aligning with in vitro findings that showed slow hepatic metabolism and limited metabolic turnover in human and animal hepatocytes, except in dogs [31]. Similar metabolic patterns were reported in dog and human microsomes, involving mono-oxidation and hydrolysis, with no evidence of reactive metabolite formation in any species [31].

While these data support low clearance with limited renal elimination, the elimination pathways of DNDI-6148 remain incompletely understood. The apparent decline in clearance and increase in half-life at higher doses may reflect saturation of hepatic processes, but other factors could contribute. As DNDI-6148 was administered as an oral suspension, it cannot be excluded that dose-dependent processes during the absorption phase contributed to the apparent decrease in clearance at higher doses. However, flip-flop kinetics is uncommon and typically associated with low-permeability BDDCS Class 3 or 4 compounds [44].

In contrast to our population PK analysis, the noncompartmental analysis reported similar estimates for CL/F but a dose-dependent increase in apparent V/F, ranging from 53 to 92 L [37]. A covariate effect of dose on V/F was also statistically significant in the population model. Models including dose effects on F and CL/F provided comparable fits to those with dose effects on V/F alone. However, preclinical studies did not indicate saturation of tissue binding; for instance, blood-to-plasma ratios remained close to 1 across a wide range of concentrations (1, 10, and 100 µM). Given the evidence for solubility-limited absorption, we therefore considered a dose effect on F, together with a dose effect on CL/F, to be more physiologically plausible.

Dose effects on both CL/F and F were described using exponential functions centered on the median dose, selected as a conservative and physiologically reasonable approximation after evaluation of linear, power, and $E_{MAX}$ relationships. Although $E_{MAX}$ models yielded marginally better fits, they were not retained due to high parameter uncertainty and added model complexity. Extrapolation beyond the studied dose range should be approached with caution due to the nature of the exponential function.

Overall, this analysis confirms DNDI-6148's favorable PK profile in healthy participants, with good permeability, moderate tissue distribution, low clearance, reasonable half-life, and minimal circulating metabolites.

DNDI-6148 was well tolerated in this FIH study at single ascending doses up to 380 mg [37]. While it has shown promising antitrypanosomal activity in preclinical murine and hamster models [31–33], its clinical efficacy remains to be demonstrated. The developed population PK model has been used to support translational analyses and dose-finding simulations; results will be reported separately.

## Strengths and limitations

A major strength of this study was the application of nonlinear mixed-effects modeling, which enabled a pooled analysis across all dose groups. This approach supported the quantification of interindividual variability and model-based exploration of key covariate effects with greater statistical power than noncompartmental analysis.

However, the study has several limitations. It was conducted in healthy male White participants, potentially limiting generalizability to target populations with CD or Leishmaniasis, who may exhibit greater demographic and PK variability. Food effects remain unstudied but should be investigated in future trials due to their potential impact on absorption. The present analysis is based on oral administration only. While the population PK model adequately described the observed data, intravenous administration would be required to fully disentangle absorption-related from elimination-related processes. Further research, including mass balance studies, is needed to fully elucidate the metabolic pathways of DNDI-6148. Finally, the assessment of potential for drug-drug interactions would be valuable, in the context of combination therapies for VL and CD.

## Conclusions

In summary, we have characterized the population pharmacokinetic properties of DNDI-6148 in this FIH study. The developed model adequately captured its nonlinear pharmacokinetics and provides a valuable tool to inform dose finding for future clinical trials. The favourable pharmacokinetic and safety profile of DNDI-6148 supports its continued development as a promising oral treatment for leishmaniasis and Chagas disease. An effective oral therapy would significantly enhance access to care, particularly in resource-limited settings. Future studies in patients are urgently needed to confirm its safety, efficacy, and therapeutic potential in target populations.

## Supporting information

**S1 Fig. Covariate effects of dose on DNDI-6148 pharmacokinetic parameters.**
(DOCX)

**S2 Fig. Distribution of individual primary and secondary plasma PK parameters of DNDI-6148 by dose level.**
(DOCX)

**S1 Code. NONMEM code of the final population PK model.**
(DOCX)

## Acknowledgments

The authors sincerely thank all participants for their participation in this study with DNDI-6148. We are also grateful to the staff at Eurofins Optimed, Gières (France), for collecting and curating the study data.

## Author contributions

**Conceptualization:** Richard M. Hoglund, Charles E. Mowbray, Jean-Yves Gillon, Eric Chatelain, Ivan Scandale, Joel Tarning.

**Data curation:** Frauke Assmus.

**Formal analysis:** Frauke Assmus, Ayorinde Adehin, Richard M. Hoglund, Joel Tarning.

**Funding acquisition:** Ivan Scandale, Joel Tarning.

**Investigation:** Frauke Assmus, Ayorinde Adehin, Richard M. Hoglund, Joel Tarning.

**Methodology:** Frauke Assmus, Ayorinde Adehin, Richard M. Hoglund, Joel Tarning.

**Project administration:** Jean-Yves Gillon, Stéphanie Braillard, Eric Chatelain, Ivan Scandale, Joel Tarning.

**Resources:** Ivan Scandale, Joel Tarning.

**Supervision:** Richard M. Hoglund, Ivan Scandale, Joel Tarning.

**Validation:** Frauke Assmus.

**Visualization:** Frauke Assmus.

**Writing – original draft:** Frauke Assmus.

**Writing – review & editing:** Frauke Assmus, Ayorinde Adehin, Richard M. Hoglund, Charles E. Mowbray, Jean-Yves Gillon, Séverine Blesson, Stéphanie Braillard, Eric Chatelain, Ivan Scandale, Joel Tarning.

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
