## [Decision Letter · Decision Letter 0]

16 Dec 2025

PNTD-D-25-01980

Towards New Treatments for Leishmaniasis and Chagas Disease: Population Pharmacokinetics of DNDI-6148 in Healthy Adults

Dear Dr. Tarning,

Thank you for submitting your manuscript to PLOS Neglected Tropical Diseases. After careful consideration, we feel that it has merit but does not fully meet PLOS Neglected Tropical Diseases's publication criteria as it currently stands. Therefore, we invite you to submit a revised version of the manuscript that addresses the points raised during the review process.

We look forward to receiving your revised manuscript.

Kind regards,

Mitali Chatterjee

Academic Editor

Abhay Satoskar

Section Editor

Shaden Kamhawi

co-Editor-in-Chief

Paul Brindley

co-Editor-in-Chief

**Journal Requirements:**

At this stage, the following Authors/Authors require contributions: Ivan Scandale. Please ensure that the full contributions of each author are acknowledged in the Add/Edit/Remove Authors section of our submission form.

4) We do not publish any copyright or trademark symbols that usually accompany proprietary names, eg ©,  ®, or TM  (e.g. next to drug or reagent names). Therefore please remove all instances of trademark/copyright symbols throughout the text, including:

- ® on pages: 5, and 22.

5) Please upload all main figures as separate Figure files in .tif or .eps format. For more information about how to convert and format your figure files please see our guidelines:

6) We note that you have indicated that there are restrictions to data sharing for this study. PLOS only allows data to be available upon request if there are legal or ethical restrictions on sharing data publicly. For more information on unacceptable data access restrictions, please see https://journals.plos.org/plosntds/s/data-availability#loc-unacceptable-data-access-restrictions.

b) If there are no restrictions, please upload the minimal anonymized data set necessary to replicate your study findings to a stable, public repository and provide us with the relevant URLs, DOIs, or accession numbers. For a list of recommended repositories, please see https://journals.plos.org/plosone/s/recommended-repositories. You also have the option of uploading the data as Supporting Information files, but we would recommend depositing data directly to a data repository if possible.

**Reviewers' comments:**

Reviewer's Responses to Questions

**Key Review Criteria Required for Acceptance?**

**Methods**

-Are the objectives of the study clearly articulated with a clear testable hypothesis stated?

-Is the study design appropriate to address the stated objectives?

-Is the population clearly described and appropriate for the hypothesis being tested?

-Is the sample size sufficient to ensure adequate power to address the hypothesis being tested?

-Were correct statistical analysis used to support conclusions?

-Are there concerns about ethical or regulatory requirements being met?

Reviewer #1: (No Response)

Reviewer #2: The study design and data analysis approaches are well described.

**Results**

-Does the analysis presented match the analysis plan?

-Are the results clearly and completely presented?

-Are the figures (Tables, Images) of sufficient quality for clarity?

Reviewer #1: (No Response)

Reviewer #2: The applied data analyses are sound and limitations appropriately discussed. Data is well presented in Figures and Tables.

**Conclusions**

-Are the conclusions supported by the data presented?

-Are the limitations of analysis clearly described?

-Do the authors discuss how these data can be helpful to advance our understanding of the topic under study?

-Is public health relevance addressed?

Reviewer #1: (No Response)

Reviewer #2: All data is appropriately discussed. The article provides a relevant resource for further DNDI-6148 development and puts the data into context for non-expert readers.

**Editorial and Data Presentation Modifications?**

Reviewer #1: (No Response)

Reviewer #2: (No Response)

**Summary and General Comments**

Reviewer #1: Assmus and colleagues have fitted a population PK model to previously published healthy volunteer data on the PK of DNDI-6148. Finding dose proportional AUC but decreasing Cmax they fit a model with dose decreasing bioavailability, offset with decreasing CL/F. The paper is clearly written and easy to follow and the model seems to describe the data well with precisely-estimated parameters. Authors should consider the following:

Title could be shortened to: Population Pharmacokinetics of DNDI-6148 in Healthy Adults

Line 120: please provide assay lower limit of quantification and precision.

Line 141: please justify fixing %CV values <10% to zero with a reference.

Line 148: please write out the exact dose function. Others have used mg/m2 or allometric 0.75 since this is more likely to scale with gastric surface area and extrapolate to special populations e.g. children. Please also justify scaling with linear body weight.

Line 152: Are authors saying they tested each covariate three times as a linear, exponential and power? What is the justification for this e.g. the power and exponential can both approximate linear and each other.

Line 209: Unclear why two equations required given only oral data used? How are beta_dose_F and beta_dose_CL structurally identifiable?

Fig 2: GOF plots require a smooth and CWRES dashed lines at -2 and +2

Table 2: put the equations for the two beta terms into the footnote to aid interpretation of parameters.

A further figure showing how typical F and CL/F change with dose i.e. drawing out the two functions, would help readers interpret the way authors have chosen to handle the nonlinearity

Reviewer #2: Assmus et al. carried out a population pharmacokinetics study for the benzoxaborole DNDI-6148, currently in the DNDi development pipeline for treatment of visceral leishmaniasis. Additionally, the compound was suggested as a candidate drug for cutaneous leishmaniasis (PMID: 30922847) and Chagas disease (PMID: 34711050). The study builds on a FIH Phase 1 study in which DNDI-6148 was shown to be safe and well tolerated after a single oral dose.

The outcome of the population pharmacokinetics study with 48 healthy participants revealed non-linear dose–exposure relationship and will be a valuable resource for clinical trials.

PLOS authors have the option to publish the peer review history of their article (what does this mean?). If published, this will include your full peer review and any attached files.

Reviewer #1: No

Reviewer #2: **Yes:**Martin Zoltner

**Figure resubmission:**

After uploading your figures to PLOS’s NAAS tool - https://ngplosjournals.pagemajik.ai/artanalysis, NAAS will process the files provided and display the results in the Uploaded Files section of the page as the processing is complete. If the uploaded figures meet our requirements (or NAAS is able to fix the files to meet our requirements), the figure will be marked as fixed above. If NAAS is unable to fix the files, a red failed label will appear above. When NAAS has confirmed that the figure files meet our requirements, please download the file via the download option, and include these NAAS processed figure files when submitting your revised manuscript.
---

## [Decision Letter · Decision Letter 1]

8 Feb 2026

PNTD-D-25-01980R1

Population Pharmacokinetics of DNDI-6148 in Healthy Adults

Dear Dr. Tarning,

Thank you for submitting your manuscript to PLOS Neglected Tropical Diseases. After careful consideration, we feel that it has merit but does not fully meet PLOS Neglected Tropical Diseases's publication criteria as it currently stands. Therefore, we invite you to submit a revised version of the manuscript that addresses the points raised during the review process.

Please submit your revised manuscript within by Mar 10 2026 11:59PM. If you will need more time than this to complete your revisions, please reply to this message or contact the journal office at plosntds@plos.org. Please include the following items when submitting your revised manuscript:

We look forward to receiving your revised manuscript.

Kind regards,

Mitali Chatterjee

Academic Editor

Abhay Satoskar

Section Editor

Shaden Kamhawi

co-Editor-in-Chief

Paul Brindley

co-Editor-in-Chief

**Journal Requirements:**

**Reviewers' Comments:**

Reviewer's Responses to Questions

**Key Review Criteria Required for Acceptance?**

**Methods**

-Are the objectives of the study clearly articulated with a clear testable hypothesis stated?

-Is the study design appropriate to address the stated objectives?

-Is the population clearly described and appropriate for the hypothesis being tested?

-Is the sample size sufficient to ensure adequate power to address the hypothesis being tested?

-Were correct statistical analysis used to support conclusions?

-Are there concerns about ethical or regulatory requirements being met?

Reviewer #1: yes

**Results**

-Does the analysis presented match the analysis plan?

-Are the results clearly and completely presented?

-Are the figures (Tables, Images) of sufficient quality for clarity?

Reviewer #1: yes

**Conclusions**

-Are the conclusions supported by the data presented?

-Are the limitations of analysis clearly described?

-Do the authors discuss how these data can be helpful to advance our understanding of the topic under study?

-Is public health relevance addressed?

Reviewer #1: not quite

**Editorial and Data Presentation Modifications?**

Reviewer #1: no

**Summary and General Comments**

Reviewer #1: Authors have mostly answered my comments in a satisfactory manner but the issue remains on the scaling of F by linear body weight mg/kg. If the model is to be extrapolated across wider size ranges than those studied here, it is physiologically more plausible to scale by BSA (e.g. see https://pubmed.ncbi.nlm.nih.gov/33560094/) because gastric surface area scales with BSA. Whilst authors are unlikely to see a radically different fit with this model in their volunteer data, if the model is to be used to extrapolate e.g. to children, it is important covariates are mechanistically more plausible.

PLOS authors have the option to publish the peer review history of their article (what does this mean?). If published, this will include your full peer review and any attached files.

Reviewer #1: No

**Figure resubmission:**

After uploading your figures to PLOS’s NAAS tool - https://ngplosjournals.pagemajik.ai/artanalysis, NAAS will process the files provided and display the results in the Uploaded Files section of the page as the processing is complete. If the uploaded figures meet our requirements (or NAAS is able to fix the files to meet our requirements), the figure will be marked as fixed above. If NAAS is unable to fix the files, a red failed label will appear above. When NAAS has confirmed that the figure files meet our requirements, please download the file via the download option, and include these NAAS processed figure files when submitting your revised manuscript.
---

## [Decision Letter · Decision Letter 2]

6 Apr 2026

Dear Professor Tarning,

We are pleased to inform you that your manuscript 'Population Pharmacokinetics of DNDI-6148 in Healthy Adults' has been provisionally accepted for publication in PLOS Neglected Tropical Diseases.

Best regards,

Mitali Chatterjee

Academic Editor

Abhay Satoskar

Section Editor

Shaden Kamhawi

co-Editor-in-Chief

Paul Brindley

co-Editor-in-Chief

Reviewer's Responses to Questions

**Key Review Criteria Required for Acceptance?**

**Methods**

-Are the objectives of the study clearly articulated with a clear testable hypothesis stated?

-Is the study design appropriate to address the stated objectives?

-Is the population clearly described and appropriate for the hypothesis being tested?

-Is the sample size sufficient to ensure adequate power to address the hypothesis being tested?

-Were correct statistical analysis used to support conclusions?

-Are there concerns about ethical or regulatory requirements being met?

Reviewer #1: (No Response)

**Results**

-Does the analysis presented match the analysis plan?

-Are the results clearly and completely presented?

-Are the figures (Tables, Images) of sufficient quality for clarity?

Reviewer #1: (No Response)

**Conclusions**

-Are the conclusions supported by the data presented?

-Are the limitations of analysis clearly described?

-Do the authors discuss how these data can be helpful to advance our understanding of the topic under study?

-Is public health relevance addressed?

Reviewer #1: (No Response)

**Editorial and Data Presentation Modifications?**

Reviewer #1: (No Response)

**Summary and General Comments**

Reviewer #1: (No Response)

PLOS authors have the option to publish the peer review history of their article (what does this mean?). If published, this will include your full peer review and any attached files.

Reviewer #1: No

---

## [Editor Report · Acceptance letter]

Dear Professor Tarning,

We are delighted to inform you that your manuscript, "Population Pharmacokinetics of DNDI-6148 in Healthy Adults," has been formally accepted for publication in PLOS Neglected Tropical Diseases.

Best regards,

Shaden Kamhawi

co-Editor-in-Chief

Paul Brindley

co-Editor-in-Chief
